# Anti-*Toxoplasma gondii* Effects of Lipopeptide Derivatives of Lycosin-I

**DOI:** 10.3390/toxins15080477

**Published:** 2023-07-26

**Authors:** Xiaohua Liu, Peng Zhang, Yuan Liu, Jing Li, Dongqian Yang, Zhonghua Liu, Liping Jiang

**Affiliations:** 1Department of Parasitology, Xiangya School of Medicine, Central South University, Changsha 410013, China; 226501025@csu.edu.cn (X.L.); liuyuan@gdim.cn (Y.L.); lijing0807@csu.edu.cn (J.L.); yangdongqian1120@csu.edu.cn (D.Y.); 2The National & Local Joint Engineering Laboratory of Animal Peptide Drug Development, College of Life Sciences, Hunan Normal University, Changsha 410081, China; pengzhang@hunnu.edu.cn (P.Z.); liuzh@hunnu.edu.cn (Z.L.); 3China-Africa Research Center of Infectious Diseases, Xiangya School of Medicine, Central South University, Changsha 410013, China

**Keywords:** fatty acid chain modification, Lycosin-I, lipopeptide, *Toxoplasma gondii*

## Abstract

Toxoplasmosis, caused by *Toxoplasma gondii* (*T. gondii*), is a serious zoonotic parasitic disease. We previously found that Lycosin-I exhibited anti-*T. gondii* activity, but its serum stability was not good enough. In this study, we aimed to improve the stability and activity of Lycosin-I through fatty acid chain modification, so as to find a better anti-*T. gondii* drug candidate. The α/ε-amino residues of different lysine residues of Lycosin-I were covalently coupled with lauric acid to obtain eight lipopeptides, namely L-C_12_, L-C_12_-1, L-C_12_-2, L-C_12_-3, L-C_12_-4, L-C_12_-5, L-C_12_-6, and L-C_12_-7. Among these eight lipopeptides, L-C_12_ showed the best activity against *T. gondii* in vitro in a trypan blue assay. We then conjugated a shorter length fatty chain, aminocaproic acid, at the same modification site of L-C_12_, namely L-an. The anti-*T. gondii* effects of Lycosin-I, L-C_12_ and L-an were evaluated via an invasion assay, proliferation assay and plaque assay in vitro. A mouse model acutely infected with *T. gondii* tachyzoites was established to evaluate their efficacy in vivo. The serum stability of L-C_12_ and L-an was improved, and they showed comparable or even better activity than Lycosin-I did in inhibiting the invasion and proliferation of tachyzoites. L-an effectively prolonged the survival time of mice acutely infected with *T. gondii*. These results suggest that appropriate fatty acid chain modification can improve serum stability and enhance anti-*T. gondii* effect of Lycosin-I. The lipopeptide derivatives of Lycosin-I have potential as a novel anti-*T. gondii* drug candidate.

## 1. Introduction

*Toxoplasma gondii* (*T. gondii*), a unicellular eukaryote, is a serious pathogenic organism that parasitizes the nucleated cells of humans and various warm-blooded animals [1]. Toxoplasmosis caused by *T. gondii* is a zoonotic parasitic disease. Approximately one-third of people worldwide are infected with *T. gondii* [2]. Domestic and wild animals are also at risk of toxoplasmosis. Approximately one-third of domestic and non-domestic cats have been exposed to *T. gondii* [3]. Although acute infection with *T. gondii* is usually asymptomatic in immunocompetent individuals, or in rare cases may cause a mild flu-like illness, the parasite typically remains active in immunocompromised individuals, leading to persistent host cell infection and acute disease [4,5,6,7].

Early research into anti-*T. gondii* drugs began in the 1940s. In the early 1950s, Eyles and Coleman observed the synergistic effect of the sulfadiazine–pyrimethamine combination in treating toxoplasmosis in mice [8]. More than 60 years later, the sulfadiazine–pyrimethamine (SDZ/PYR) combination remains the gold standard for treating toxoplasmosis in humans. However, these drugs have several drawbacks, including a long treatment duration, a high relapse rate and significant adverse effects. In addition, there are currently no drugs available to remove the cysts and destroy the bradyzoites within cysts [9,10]. Researchers have made unremitting efforts to find novel anti-*T. gondii* drugs with better activity, higher stability, and fewer side effects, focusing mainly on the following areas: (1) To discover the new function of anti-*T. gondii* from the current clinical drugs. For example, NVP-AEW541, GSK-J4, pravastatin, simvastatin and the clinical antibiotic enrofloxacin have been found to have anti-*T. gondii* infection [11,12,13]. (2) Some compounds extracted from plants have been found to have anti-*T. gondii* activity. For example, 3-deoxyanthocyanidins extracted from sorghum bicolor shows anti-*T. gondii* activity in vitro [14]. The aqueous extracts of Astragalus membranaceus and Scutellaria baicalensis also show potent anti-*T. gondii* activity [15]. Ursolic acid, found in various medicinal plants, can induce the direct inhibition of *T. gondii* or increase the survival of mice by effectively blocking and inhibiting the viability of *T. gondii* [16]. (3) Compounds extracted from animal natural products [17], such as Bplec, a C-type lectin, and BNSP-7, a phospholipase analogue, isolated from the crude venom of *Bothrops Pauloensis*, have anti-*T. gondii* activity [18,19]. (4) Peptides. In recent years, peptides have played an important role in key research areas such as pharmacology, physiology and neurobiology [20]. To date, only a few peptides with anti-*T. gondii* activity have been reported in the literature. The 17-amino acid peptide, cal14.1a, produced by *Conus californicus*, reduces the invasion and proliferation of *T. gondii* [21]. Longicin P4, a 21 amino acid basic peptide from *Haemaphysalis longicornis*, has an inhibitory effect on *T. gondii*, Gram-negative and Gram-positive bacteria [22]. In addition, our previous studies have shown that the peptide Lycosin-I from the spider *Lycosa singoriensis*, the crude venoms from the spiders *Ornitoctonus huwena* and *Chilobrachys jingzhao*, and the peptide XYP1 from the spider *Lycosa coelestis* have significant anti-*T. gondii* activity in vitro and in vivo [23,24,25].

Cationic peptides with an α-helical structure found in venoms constitute a class of antimicrobial peptides (AMPs). In this class, amphiphilic peptides with net positive charges tend to adopt an α-helix structure when in contact with a hydrophilic/hydrophobic interface (such as the membrane of microorganisms) [26], which helps the AMPs to insert into the cell membrane and disrupt the integrity of the cell membrane [27]. Phospholipids on cell membranes are one of the targets of transmembrane peptides with an α -helical structure [28]. As a member of the AMP family, Lycosin-I plays a bactericidal role by combining with the bacterial cell membrane through electrostatic action to disrupt the membrane structure and cause the contents to leak out [29,30]. It is also a cell-penetrating peptide (CPP) that can carry coupled spherical gold nanoparticles into cancer cells [31]. However, the serum stability of Lycosin-I is not good enough, and it is completely degraded into smaller peptides after incubation in mouse serum for 24 h [32].

To improve the structural stability and activity of the peptide, its structure is modified via various methods including acetylation, amidation, amino acid replacement, D-amino acid substitution, the deletion of inactive amino acids, cyclization into a cyclic peptide, and fatty acid modification into a lipopeptide. Recent studies have shown that cyclic peptides and lipopeptides can enhance the biological activity and stability of peptides [32,33,34,35]. Fatty acids, a class of essential nutrients, regulate a wide range of vital processes and disorders including diabetes, cancer, and cardiovascular disease [36,37]. The fatty acid modification of peptides has high hydrophobicty and durability, making it a very promising method for improving peptide function. Glucagon-like peptide-1 analogs from *Xenopus laevis* modified with mycophenolic acid analogs containing fatty acids of different lengths (aminocaproic acid and dodecanoic acid) significantly improved the efficacy and duration of anti-diabetes treatment [38].

Fatty acids are carboxylic acids with long aliphatic chains that bind to the N-terminus or lysine side chains in peptides. In addition, cysteine residues can also be modified by fatty acids to obtain corresponding lipopeptide derivatives. Peptides modified with fatty acids are mainly used for the following aspects: (1) To improve the transmission of the molecular peptides mediated [39,40]. PepFect14 combined with acyl chains ranging in length from 2 to 22 carbons can optimize the minimal toxicity of low-nucleotide complexes and maintain delivery efficiency [41]. (2) To increase peptide stability. Myristic acid binds to lysine side chains on the insulin alpha chain to provide sufficient stability [42]. In addition, the antibacterial peptide analogs modified with fatty acids at the N-terminus showed high stability in the presence of protease or serum [43]. (3) Enhancing the activity of peptides. For example, B1 coupled with different lengths of fatty acids results in better anti-tumor activity [44]. The N-terminus of Lycosin-I was coupled with fatty acids of different lengths, demonstrating how fatty acids can increase the activity of peptides and how the activity of the peptides that do so differ, with lauric acid showing the most improved activity [45]. Subsequently, the lipopeptides L-C_12_, L-C_12_-1, L-C_12_-2, L-C_12_-3, L-C_12_-4, L-C_12_-5, L-C_12_-6 and L-C_12_-7 were obtained via the covalent coupling of lauric acid with α/ε -amino groups of lysine residues at different positions. Their biological properties, such as cytotoxicity and anti-tumor cell metastasis, as well as their particle size, zeta potential, secondary structure, hydrophobicity, and serum stability were identified. L-C_12_ performed the best among these lipopeptides and its serum stability was significantly increased compared to that of Lycosin-I [32].

Our previous studies confirmed that the spider peptide Lycosin-I has anti-*T. gondii* activity [23]. In this study, we believe that the position of the fatty acid chain modification and the length of the fatty chain may be the key factors for peptide optimization, so we will further investigate the effect of fatty acid modification with different fatty chain lengths (lauric acid and aminocaproic acid) on the anti-*T. gondii* activity of Lycosin-I.

## 2. Results

### 2.1. The Anti-T. gondii Efficacy of Lipopeptides

Eight lipopeptides, covalent couplings of lauric acid with α/ε-amino groups of lysine residues at different sites of Lycosin-I, were synthesized according to the preparation route [32] and designated as L-C_12_, L-C_12_-1, L-C_12_-2, L-C_12_-3, L-C_12_-4, L-C_12_-5, L-C_12_-6, and L-C_12_-7 (Figure 1A). The secondary structure of Lycosin-I shows that all the lysines are in an α-helical structure except for the first site where the lysine is located on the linear structure (Figure 1B). As our previous studies confirmed that Lycosin-I has concentration-dependent anti-*T. gondii* activity, we set up different concentration gradients (2.5 μM, 5 μM, 10 μM, 20 μM, and 40 μM) to evaluate the activity of the lipopeptides against *T. gondii* in vitro. We estimated the mortality of tachyzoites treated with lipopeptides at different concentrations using the trypan blue assay (Appendix A). Each had a different level of activity against *T. gondii*. Among them, L-C_12_ showed the best efficacy against *T. gondii* in vitro. There was no significant difference in anti-*T. gondii* activity between L-C_12_ and Lycosin-I (Appendix A). We then coupled another fatty chain length (aminocaproic acid) to the same modification site of L-C_12_, namely L-an (Figure 1A), which also showed no statistical difference from Lycosin-I in anti-*T. gondii* activity (Appendix A). The 50% mortality rates of Lycosin-I, L-C_12_ and L-an against *T. gondii* were 7.20 μM, 6.54 μM and 7.10 μM, respectively. In addition, lauric acid and aminocaproic acid alone had no effect on *T. gondii* in vitro.

### 2.2. L-C_12_ and L-an Increased the Serum Stability of Anti-T. gondii In Vitro

A combination of serum incubation and a trypan blue assay was performed to determine whether or not the anti-*T. gondii* activity of lipopeptides was affected by serum. Compared to the control group (untreated with serum), Lycosin-I treated with serum significantly had reduced anti-*T. gondii* activity, whereas L-C_12_ and L-an showed no significant difference (Figure 2A–C).

To exclude the effect of lipopeptide toxicity on *T. gondii* activity, the effect of lipopeptides on the host cell human foreskin fibroblast (HFF) viability was evaluated using the CCK-8 assay. Lycosin-I, L-C_12_, and L-an showed concentration-dependent cytotoxicity to cells. Compared to Lycosin-I, L-C_12_ showed higher cytotoxicity after modification, while L-an showed no significant difference (Figure 2D). The half-inhibitory concentrations (IC_50_) of Lycosin-I, L-C_12_ and L-an were 46.16 μM, 15.62 μM and 30.95 μM, respectively.

### 2.3. L-C_12_ and L-an Inhibit the Invasion and Proliferation of Tachyzoites into Host Cells

The tachyzoites of *T. gondii* infect cells in a relatively rapid process. They can invade host cells after 30 min of contact, forming PVs, and a host cell can be infected with multiple tachyzoites simultaneously (Appendix A). By calculating the invasion rate of tachyzoites, we found that in the DMEM group (negative control), the invasion rate of tachyzoites on host cells reached more than 50% after 2 h. L-C_12_ and L-an had a better inhibition effect on tachyzoite invasion than Lycosin-I did especially at a low concentration of 5 μM (Figure 3A). It was not unexpected that Lycosin-I, L-C_12_ and L-an showed more pronounced inhibition than SDZ (a clinical drug for toxoplasmosis) did at the concentration of 10 μM, because SDZ targets dihydropteroate synthase, affecting *T. gondii* proliferation rather than directly killing *T. gondii*.

After invading host cells, tachyzoites proliferate exponentially in the PV and can be arranged in a typical rose shape when they multiply to a certain extent. HFFs infected with tachyzoites were incubated with Lycosin-I, L-C_12_ and L-an at concentrations of 5 μM and 10 μM for 24 h. After Giemsa staining, the PVs in HFFs were observed under a light microscope (Appendix A). By counting the proportion of PVs with different numbers of tachyzoites in 100 PVs and calculating the proliferation rate of tachyzoites, we found that after incubation for 24 h, the proliferation of tachyzoites in the host cells was significantly inhibited in all drug treatment groups compared to that in the DMEM group. The proliferation rate of tachyzoites in the HFFs treated with Lycosin-I, L-C_12_ and L-an at 10 μM decreased to below 55% (Figure 3B). In the DMEM group, the number of tachyzoites in the PV was mostly 4 or 8, and some PVs with 16 tachyzoites could be seen. In contrast to the DMEM group, all treatment groups showed some inhibition of tachyzoite proliferation in PV. The number of PVs with 8 or 16 tachyzoites decreased significantly, while the proportion of PVs with 1, 2 and 4 tachyzoites increased. It was hard to see a PV with 16 tachyzoites in Lycosin-I, L-C_12_ and L-an at the concentration of 10 μM (Figure 3C).

A plaque assay can provide a thorough representation of the ability of tachyzoites to invade, proliferate and migrate. Based on the above results, a concentration of 10 μM was chosen for the plaque assay. HFFs infected with tachyzoites were incubated with DMEM (negative control, DMEM), SDZ (a clinical drug for toxoplasmosis, 10 μM), Lycosin-I (10 μM), L-C_12_ (10 μM) and L-an (10 μM), respectively, and HFFs not infected with tachyzoites were incubated with DMEM alone (blank control group). After 7 days of incubation, there was no obvious plaque in the drug-positive SDZ group compared to the DMEM group. The plaques of the Lycosin-I, L-C_12_ and L-an groups were also reduced (Figure 3D,E). In contrast with those in the DMEM group, the number and area of the plaques in SDZ, Lycosin-I, L-C_12_ and L-an were significantly reduced (Figure 3F,G), indicating that these treatment groups effectively inhibited the proliferation and re-invasion of tachyzoites in HFFs at the concentration of 10 μM. Compared Lycosin-I, L-C_12_ and L-an showed much better anti-*T. gondii* efficacy in vitro.

### 2.4. The Efficacy of Lipopeptides on Anti-T. gondii In Vivo

We have shown that L-C_12_ and L-an exhibited the same significant anti-*T. gondii* activity as Lycosin-I did in vitro. To evaluate the efficacy of the lipopeptides in vivo, we established a mouse model of acute *T. gondii* infection. Although SDZ exerts its anti-*Toxoplasma* effect at high concentrations in other papers, and its anti-*Toxoplasma* effect was also observed at 400 µM in vitro and 100 mg/kg in vivo in our previous study [25], in this study, we wanted to explore whether or not the anti-*Toxoplasma* effect of the peptides was better at the same low concentration as that of the SDZ, so we set the SDZ at the same concentration as that of the peptides in both the in vitro and in vivo experiments. Therefore, the mice were treated with 4 mg/kg/day of SDZ or lipopetides for 7 days, and the survival time of the mice was closely monitored for 15 days. Compared to the control group, SDZ and L-C_12_ could not prolong the survival time of the mice, while Lycosin-I and L-an significantly prolonged the survival time of the mice (Figure 4A). We also determined the parasite load in the peritoneal fluid and tissues of mice acutely infected with *T. gondii* after 5 days of treatment. The number of tachyzoites in the peritoneal fluid of the mice was directly counted using a blood cell counting plate (Figure 4B), while the expression of the surface antigen SAG1, a membrane surface protein specific to tachyzoites, was detected via qRT-PCR in the heart, liver, spleen, lung and brain of the mice (Figure 4C–G). Compared to that in the PBS group (negative control), the parasite load was significantly reduced in all treated groups.

### 2.5. The Expression of Inflammatory Factors in Mice

Once *T. gondii* has infected a host, a number of inflammatory responses can be induced in the host to resist *T. gondii* invasion and proliferation. Therefore, we wanted to investigate the expression levels of anti-inflammatory and pro-inflammatory factors in the mice treated with lipopeptides. The expression levels of IFN-γ, TNF-α, IL-4 and IL-10 in the spleen tissues of mice after 5 days of treatment were determined via qRT-PCR (Figure 5). Compared to the blank control group (mice not infected with *T. gondii*), the pro-inflammatory cytokines IFN-γ and the anti-inflammatory cytokines IL-4 and IL-10 were significantly upregulated in PBS (negative control; mice infected with *T. gondii* without drug treatment). Compared to the PBS group, all treatment groups promoted the upregulation of the pro-inflammatory cytokines IFN-γ and TNF-α and the anti-inflammatory cytokine IL-4. SDZ, Lycosin-I and L-an inhibited the upregulation of IL-10, whereas L-C_12_ promoted it.

## 3. Discussion

The combination of sulfadiazine and pyrimethamine remains the gold standard for the treatment of toxoplasmosis. However, pyrimethamine is potentially teratogenic and causes reversible bone marrow suppression [9,46,47,48]. In addition, various serious complications of clinical drugs for toxoplasmosis have been reported, such as agranulocytosis, Stevens–Johnson syndrome, toxic epidermal necrolysis and hepatic necrosis, etc. [49,50,51,52,53]. Current treatment for toxoplasmosis is limited, with multiple side effects and long treatment durations (ranging from 4–6 weeks to more than 1 year) [9,54]. In addition, some intrinsic factors of *T. gondii*, such as increased drug resistance and differential drug susceptibility among different strains, also play an important role in disease progression and treatment failure [55,56,57]. Therefore, there is an urgent need to develop safer and more effective therapeutic alternatives for toxoplasmosis with fewer side effects, which depends on the growing knowledge of *Toxoplasma* pathophysiology and the discovery of promising drug targets.

In this study, the lipopeptides L-C_12_, L-C_12_-1, L-C_12_-2, L-C_12_-3, L-C_12_-4, L-C_12_-5, L-C_12_-6 and L-C_12_-7 were preliminarily evaluated for their direct anti-*T. gondii* activity in vitro using the trypan blue assay. It was found that the L-C_12_-4 and L-C_12_-6 lipopeptides completely lost their anti-*T. gondii* activity in vitro. We suggested that the lysine at these two sites was the key site for Lycosin-I to exert its anti-*T. gondii* activity, and that the lauric acid modified at these two sites altered its original physical and chemical properties. Although L-C_12_, L-C_12_-1, L-C_12_-2, L-C_12_-3, L-C_12_-5 and L-C_12_-7 all showed concentration-dependent anti-*T. gondii* activity in vitro, only L-C_12_ showed similar efficacy to that of Lycosin-I at 10 μM and a lower concentration; the other lipopeptides were obviously weaker than Lycosin-I in anti-*T. gondii* activity. The secondary structure of Lycosin-I showed that all the lysines are in the α-helical structure except for the first site where the lysine is located on the linear structure. Therefore, we speculated that the α-amino terminus of the first lysine site of Lycosin-I (the N-terminus of Lycosin-I) was the better site for modification compared to the other lysine sites, which was consistent with the previous research findings [32].

Since fatty acids of different lengths have different activities on peptides [45], we obtained the lipopeptide L-an by coupling another shorter fatty chain (aminocaproic acid) to the N-terminus of Lycosin-I. The trypan blue assay showed that the anti-*T. gondii* activity of the two lipopeptides in vitro was similar to that of Lycosin-I. However, the cytotoxicity of L-C_12_ was significantly higher than that of Lycosin-I. L-an showed no significant difference from Lycosin-I, indicating that the longer the fatty chain was, the more toxic it was to the cells. The serum stability of Lycosin-I was increased after fatty acid modification, as shown via mass spectrometry, which showed that Lycosin-I was degraded into smaller peptides after incubation with 10% serum for 24 h, whereas L-C_12_ was not degraded [32]. Interestingly, we found that Lycosin-I only slightly decreased, rather than completely lost its anti-*T. gondii* activity in vitro after 24 h of serum incubation. which suggests that although Lycosin-I was degraded into smaller peptides, the key amino acid sequences of anti-*T. gondii* were still retained in the small peptide; maybe we could optimize the specific activity of peptides by truncating them. After incubation in serum for 24 h, L-C12 and L-an retained substantial anti-*T. gondii* activity in vitro, consistent with the results of L-C12 mass spectrometry. Although L-an was not shown via mass spectrometry to be resistant to enzymatic degradation in serum, it was confirmed via trypan blue assay that L-an improved the stability of the anti-*T. gondii* activity of Lycosin-I in serum.

The invasion and pathogenicity of *T. gondii* to a host is a complex process involving parasite movement and penetration, as well as interaction and attachment with host cells [58]. The proliferation of *T. gondii* in host cells is mainly influenced by the immune response of the host cells [59,60]. SDZ targets dihydropteroate synthase and thus affects *T. gondii* proliferation rather than directly killing *T. gondii*, so the choice of SDZ as a positive control in the invasion assay was inappropriate. However, considering that these lipopeptides were obtained by modifying Lycosin-I, which has been shown to have anti-Toxoplasma activity in our previous studies, and we mainly wanted to compare the anti-Toxoplasma activity of these modified peptides with the parent peptide Lycosin-I, no additional other positive control was performed. In vitro, Lycosin-I, L-C_12_ and L-an showed significant anti-*T. gondii* activity, which was consistent with our previous results for Lycosin-I [23]. We found that two lipopeptides, L-C_12_ and L-an, showed slightly better inhibition efficiency of tachyzoites than Lycosin-I did, especially at the low concentration of 5 μM. Although the parasitophorous vacuole membrane (PVM) was mainly derived from the plasma membrane of host cells, *T. gondii* also secreted some proteins to participate in the formation of PVM. Furthermore, some proteins of *T. gondii* are even secreted and released into host cells through the PVM, mediating the host cell’s immune response and achieving the purpose of evading host immunity [61,62,63]. In the plaque assay, Lycosin-I, L-C_12_, and L-an showed no advantages over the positive drug SDZ at the same concentration, which may have been related to the time of onset and the stability of drug efficacy. In the invasion and proliferation assays, there were 5 × 10^5^ tachyzoites (MOI = 5, tachyzoite number/cell number) in each well, and the incubation time was relatively short (only 2 h or 24 h). Fast-acting peptides therefore have an advantage over slow-acting SDZ. However, only 500 tachyzoites were used in each well in the plaque assay, and the incubation time was longer than 7 days. Peptides may be degraded by serum, resulting in a less good effect than that of positive drugs. The inhibitory effect of L-C_12_ and L-an was slightly better than that of Lycosin-I, which may have beeb related to the improved serum stability of these two lipopeptides.

Experiments on animals are the most important indicator of drug efficacy. We established a mouse model acutely infected with tachyzoites, recorded the survival time of the mice and determined the parasite load in the tissues of the mice. Lycosin-I and L-an can effectively prolong the survival time of mice. Although SDZ and L-C_12_ can effectively inhibit the proliferation of tachyzoites in mice, they could not prolong the survival time of mice at the same dose. It is a fact that the clinical dosage of SDZ is 400 mg/kg, and a low dose of SDZ/PYR (20/0.5 mg/kg) failed to protect mice from dying [64], not to mention the fact that only 4 mg/kg was used in the study. As for L-C_12_, this have may been due to its excessive toxicity to mice, which resulted in the early death of the mice. The cytotoxicity of L-C_12_ was significantly higher than that of Lycosin-I in vitro, with an IC_50_ of 15.62 μM compared to that of 46.16 μM for Lycosin-I.

*Toxoplasma* infection can induce a host immune response that is primarily mediated by T helper cell 1 (Th1) and requires components of both the innate and adaptive immune response [65]. In the initial stage of infection, *T. gondii* is recognized by innate immune response cells, stimulating dendritic cells, macrophages and neutrophils to produce interleukin-12 (IL-12) and inducing natural killer (NK) cells to produce interferon-gamma (IFN-γ). In addition, tumor necrosis factor (TNF) can interact synergistically with IL-12 to optimize IFN-γ production in these cells [66,67]. These pro-inflammatory cytokines are produced when the adaptive immune response, mediated by CD4^+^ and CD8^+^ T lymphocytes, is activated. These lymphocytes produce and secrete several inflammatory mediators, such as nitric oxide (NO), and also cause a greater increase in IL-12 and IFN-γ levels [68,69]. IFN-γ can activate dendritic cells, macrophages, and neutrophils to promote a reduction in or the elimination of *T. gondii*. When macrophages are activated by IFN-γ, the production of NO increases, contributing to the toxicity of *T. gondii* [70]. In this study, we found that IFN-γ, but not TNF-α, was significantly increased in mice acutely infected with tachyzoites compared to with normal mice. SDZ, Lycosin-I, L-C_12_ and L-an promoted the expression of IFN-γ and TNF-α in mice. However, an excessive inflammatory response can lead to host death [66,71]. Therefore, it is important to strike a balance between Th1- and Th2-mediated immune responses. This balance can be mediated by the production of anti-inflammatory cytokines, such as IL-4, IL-10, and transforming growth factor (TGF-β1), which play a role by reducing the production of NO in macrophages and the cytotoxic activity of NK cells [72,73]. Additionally, *Toxoplasma* seeks mechanisms to evade the strong immune response of a host, such as the induction of anti-inflammatory cytokines. The expression of IL-10 and TGF-β1 increased the susceptibility of BeWo cells to *T. gondii* infection. *Toxoplasma* infection can induce the production of IL-4, IL-10 and TGF-β1 in host cells [74]. In this study, we found that the expression of IL-4 and IL-10 was higher in mice acutely infected with tachyzoites than in normal mice. Interestingly, SDZ, Lycosin-I, L-C_12_ and L-an promoted IL-4 expression in mice compared to that in the negative control mice. However, SDZ, Lycosin-I and L-an inhibited the expression of IL-10, while L-C_12_ promoted the expression of IL-10. We hypothesized that the increase in IL-4 may have regulated the inflammation caused by the expression of the pro-inflammatory factors IFN-γ and TNF-α, and prevented excessive inflammation leading to the death of the mice. The reduction in IL-10 may have been related to the inhibition of tachyzoites to evade host immunity by inducing the expression of anti-inflammatory factors. As for the promotion of IL-10 expression by L-C_12_, this may have also been the reason why L-C_12_ did not prolong the survival time of the mice in the animal survival experiments, because it only focuses on promoting the expression of anti-inflammatory factors to regulate the inflammatory response, which is precisely what helps the tachyzoites to escape the host’s immune response.

## 4. Conclusions

We identified a better modification site to improve the anti-*T. gondii* activity of Lycosin-I, namely, the α-amino terminus of the first lysine site of Lycosin-I (the N-terminus of Lycosin-I). L-C_12_ and L-an obtained via the fatty acid chain modification of Lycosin-I improved serum stability. L-C_12_ and L-an showed comparable or even better activity than Lycosin-I did in inhibiting tachyzoite invasion, proliferation and migration. L-an effectively prolonged the survival time of mice acutely infected with tachyzoites. These results suggest that the lipopeptide derivative of Lycosin-I has the potential to be a novel drug candidate of anti-*T. gondii*, which expands the understanding of the effects of peptides and further enriches the research on anti-*T. gondii* drugs.

## 5. Materials and Methods

### 5.1. Animals, Cells and Parasites

Specific pathogen-free (SPF) 8-week-old female BALB/c mice and Kunming (KM) mice were obtained from the Department of Laboratory Animals, Central South University. Human foreskin fibroblasts (HFFs) were cultured in Dulbecco’s modified Eagle’s medium (DMEM) containing 10% fetal bovine serum (FBS) and 1% antibiotics (10,000 U/mL penicillin and 10 mg/mL streptomycin solution). The tachyzoites used in this study were type I of the RH strain of *T. gondii*.

### 5.2. Lipopeptide Synthesis

The secondary structure of Lycosin-I was predicted using the I-TASSER server [75] and visualized using pymol software. Lipopetides were synthesized via solid-phase peptide synthesis (SPPS) methods and purified via reversed-phase high-performance liquid chromatography (RP-HPLC) as previously described [32]. The acetonitrile gradient of RP-HPLC can be seen in Table 1. Lycosin-I and L-C_12_ were synthesized by the Jing Peptide Biotechnology Co., Ltd., Hefei, China, and Pepmic Co., Ltd., Suzhou, China, respectively. Sulfadiazine (SDZ) was purchased from Sangong Biotech, Shanghai, China. Lycosin-I and lipopeptides were dissolved in phosphate-buffered saline (PBS).

### 5.3. Lipopeptide Screening

To evaluate the difference of anti-*T. gondii* in the vitro between lipopeptides and Lycosin-I, we set up different concentration gradients, including 2.5 μM, 5 μM, 10 μM, 20 μM and 40 μM. The freshly obtained tachyzoites (3 × 10^6^) were incubated with different concentrations of lipopeptides and PBS at room temperature for 2 h, then centrifuged at 3000 rbp for 8 min and the supernatant was discarded. The tachyzoite pellet was resuspended in PBS and mixed well. Then, the tachyzoite suspension was stained with 0.4% trypan blue staining solution for 5 min. Tachyzoite viability was observed under a light microscope (Motic China Group Co., Ltd., Xiamen, China), and five fields were randomly selected to calculate tachyzoite mortality. Three independent experiments were carried out.

### 5.4. Serum Stability Assay

It has been shown via mass spectrometry that Lycosin-I but not L-C_12_ is degraded into smaller peptides after incubation in 10% serum for 24 h, indicating that Lycosin-I improves serum stability after fatty acid modification [32]. A combination of serum incubation and a trypan blue assay was used to investigate whether or not the anti-*Toxoplasma* activity of lipopeptides and Lycosin-I was affected by serum. Briefly, stock solutions (1 mM) of Lycosin-I, L-C_12_ and L-an were diluted into 100 μM solutions in PBS containing 10% fetal bovine serum, and incubated at 37 °C for 24 h. Fresh tachyzoites were then treated with these solutions as described in the method in Section 5.3 for lipopeptide screening. Three independent experiments were carried out.

### 5.5. Cell Viability Assay

HFF cells were seeded in 96-well plates (1 × 10^4^ cells per well). After 24 h of incubation, the previous complete medium was removed and replaced with the new complete medium containing different concentrations (80–1.25 μM, twofold serial dilution) of lipopeptides. Cells were treated with the complete medium in one well as the control group (each group was set up with three wells). After 24 h, 10 μL of CCK-8 reagent was added to each well and then incubated for 2 h. The absorbance value was measured at 450 nm using a microplate reader. Three independent experiments were carried out.

### 5.6. Invasion Assay

The HFF cells were seeded in 24-well plates with 14 mm coverslips (1 × 10^5^ cells per well) and incubated for 24 h. Fresh tachyzoites were obtained aseptically from the peritoneal fluid of the KM mouse acutely infected with *T. gondii*. The tachyzoites were treated with Lycosin-I (5 μM, 10 μM), L-C_12_ (5 μM, 10 μM), L-an (5 μM, 10 μM), SDZ (10 μM, positive control) and DMEM (negative control) for 2 h each. The HFFs were then infected with pre-treated tachyzoites at a multiplicity of infection (MOI) of 5 (tachyzoites/cells = 5). After 2 h, the previous complete medium was removed and the HFFs were washed twice with PBS to remove extracellular tachyzoites. They were then fixed with methanol for 5 min, and then stained with Giemsa for 15 min. The coverslips were examined under a light microscope. The invasion rate of tachyzoites was calculated by counting the number of cells invaded by tachyzoites and the total number of cells in 5 randomly selected fields. Three independent experiments were carried out.

### 5.7. Intracellular Proliferation Assay

HFFs were seeded in 24-well plates (1 × 10^5^ cells per well) with 14 mm coverslips and incubated for 24 h. The HFFs were infected with fresh tachyzoites at a MOI of 5 for 2 h. The HFFs were then washed twice with PBS to remove extracellular tachyzoites and then treated with Lycosin-I (5 μM, 10 μM), L-C_12_ (5 μM, 10 μM), L-an (5 μM, 10 μM), SDZ (10 μM, positive control) and DMEM (negative control) for 24 h each. The HFFs were then washed twice with PBS, fixed with methanol for 5 min, and then stained with Giemsa for 15 min. The coverslips were observed under a light microscope to count the number of tachyzoites in 100 parasitophorous vacuoles (PVs). Three independent experiments were carried out.

### 5.8. Plaque Assay

The HFF cells were seeded in 6-well plates (1 × 10^6^ cells per well) and incubated for 48–72 h. Cells in the 6-well plates were infected with 500 fresh tachyzoites per well for 2 h. The well in which cells were not infected with tachyzoites was considered the blank control group. Cells infected with tachyzoites were then treated with Lycosin-I (10 μM), L-C_12_ (10 μM), L-an (10 μM), SDZ (10 μM, positive control) and DMEM (negative control), respectively. After 7 days, HFFs in the plates were washed twice with PBS, fixed with methanol for 5 min, and then stained with crystal violet for 20 min. Six-well plates were washed with running water, dried, photographed, and then the number and area of plaques were counted using Photoshop version 2020. Three independent experiments were carried out.

### 5.9. Survival Assay

To evaluate the anti-*T. gondii* effect of lipopeptides in vivo, we established a mouse model acutely infected with the RH strain of *T. gondii*. Forty 8-week-old female BALB/c mice weighing 17–19 g were randomly divided into 5 groups (*n* = 8 per group) and intraperitoneally injected with 1 × 10^3^ tachyzoites per mouse. Lycosin-I, L-C_12_, L-an and sulfadiazine (SDZ) were dissolved in PBS, respectively. As these peptides are small linear peptides that are easily degraded by pepsin; mice were treated with these peptides via intraperitoneal injection (the same way that the mice were infected with *T. gondii*) rather than oral gavage. After 4 h of infection, mice were traperitoneally injected with 4 mg/kg of Lycosin-I, L-C_12_, L-an and sulfadiazine solution (once a day) for 7 days. Mice in the negative control group were treated with PBS. Mice in this condition observed daily and the time of death was recorded. The experiment was approved by the Ethics Committee of the School of Basic Medicine of Central South University (approval number: 2021-KT24).

### 5.10. RNA Extraction and Quantitative Real-Time PCR (qRT-PCR)

Total RNA was extracted from each sample using Total RNA Isolation Kit (ENOVA BIO Co., Ltd., Wuhan, China), in accordance with the manufacturer’s instructions. Reverse transcription of RNA into cDNA was performed using the PerfectStart Uni RT&qPCR Kit (TransGen Biotech Co., Ltd., Beijing, China) in accordance with the instructions. cDNA was used as a template for qRT-PCR using PerfectStart Uni RT&qPCR Kit (TransGen Biotech Co., Ltd., Beijing, China). Primers, listed in Table 2, were designed and synthesized by a biotechnology company (Sangon Biotech Co., Ltd., Shanghai, China). Analysis was performed using the comparative threshold cycle method (2^−∆∆Ct^). The gene expression levels of genes were normalized to those of housekeeping genes (β-actin or GAPDH).

### 5.11. Statistical Analysis

Data were analyzed and graphs were constructed using Photoshop version 2020 (Adobe Systems Inc., SAN Jose, CA, USA) and GraphPad Prism version 8.0 (GraphPad Software, San Diego, CA, USA). Difference statistics were calculated using the two-tailed Student’s *t*-test and *p* < 0.05 was considered statistically significant.

## Figures and Tables

**Figure 1 toxins-15-00477-f001:**
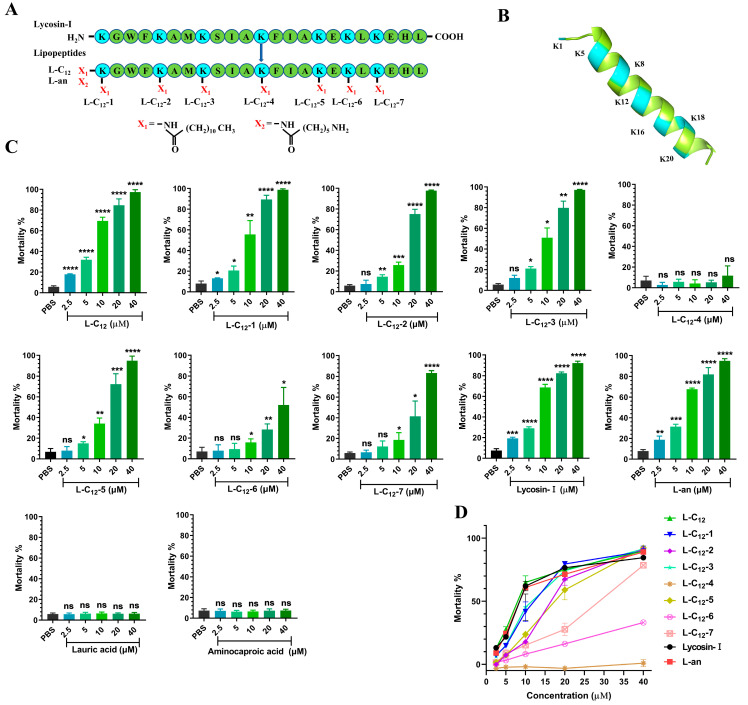
The effect of lipopeptides against *T. gondii* was evaluated using the trypan blue assay. (**A**) The modification site of the lipopeptides. Label X_1_ indicates that lauric acid was coupled to the α-amino or ε-amino group of Lys. Label X_2_ indicates that aminocaproic acid was coupled to the α-amino group of Lys. (**B**) Visualization of the secondary structure of Lycosin-I; blue indicates the lysine. (**C**) Tachyzoites treated with the control group (PBS) and different concentrations of lipopeptides. Tachyzoite viability was observed under a light microscope, and five fields were randomly selected to calculate tachyzoite mortality (ns > 0.05, * *p* < 0.05, ** *p* < 0.01, *** *p* < 0.001 and **** *p* < 0.0001 in comparison with PBS). (**D**) Figure of normalized concentration–response curves (mortality as a percentage of PBS control) constructed from the bar graphs in C.

**Figure 2 toxins-15-00477-f002:**
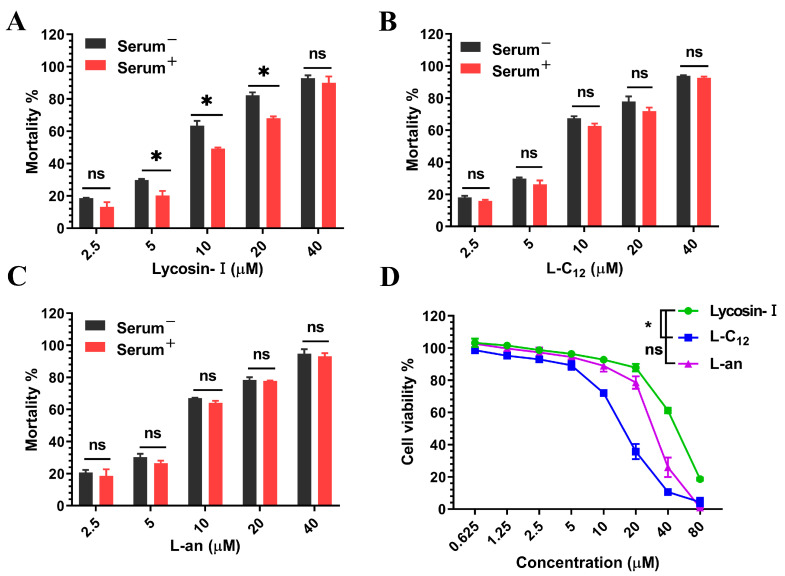
(**A**–**C**) Mortality of lipopeptides treated with serum on tachyzoites. There was a significant difference for Lycosin-I but not for L-C_12_ and L-an (* *p* < 0.05, ns > 0.05). (**D**) A CCK-8 assay was used to evaluate the toxicity of lipopeptides on HFFs (* *p* < 0.05, ns > 0.05).

**Figure 3 toxins-15-00477-f003:**
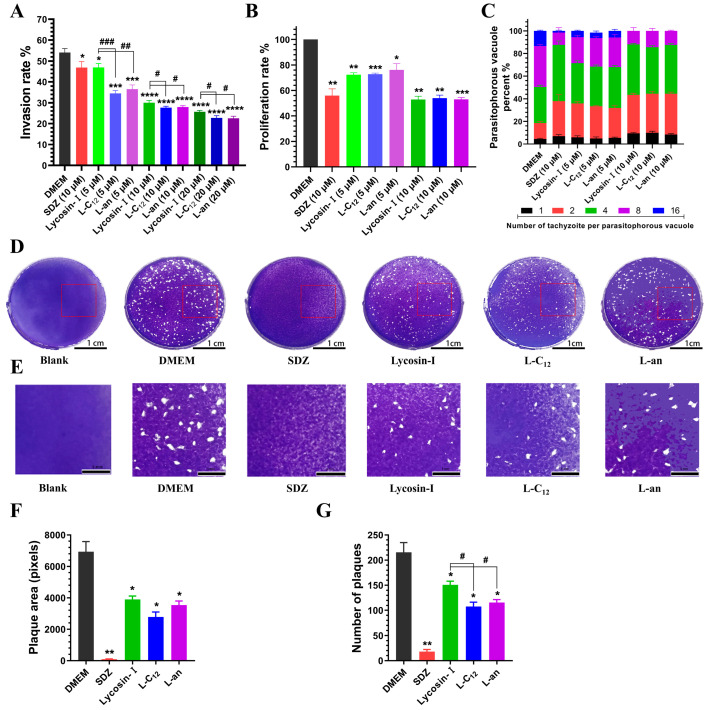
Effect of lipopeptides on invasion and proliferation of tachyzoites into host cells. (**A**) Statistical analysis of the invasion rate of tachyzoites. Tachyzoites were pretreated with DMEM (negative control), SDZ (a clinical drug for toxoplasmosis, 10 μM), Lycosin-I (5 μM and 10 μM), L-C_12_ (5 μM and 10 μM), and L-an (5 μM and 10 μM) before exposure to HFFs. (**B**) The statistics of the proliferation rate of tachyzoites (the proliferation rate of the tachyzoites in the negative group was determined as 100%). HFFs infected with tachyzoites were treated with DMEM (negative control), SDZ (a clinical drug for toxoplasmosis, 10 μM), Lycosin-I (5 μM and 10 μM), L-C_12_ (5 μM and 10 μM), and L-an (5 μM and 10 μM) for 24 h. (**C**) The proportion of PVs with different numbers of tachyzoites in 100 PVs. (**D**) Photograph of a representative well from each group of plaque assays (scale bars = 1 cm). HFFs not infected with tachyzoites were treated with DMEM (blank control). HFFs infected with tachyzoites were treated with DMEM (negative control, DMEM), SDZ (a clinical drug for toxoplasmosis, 10 μM), Lycosin-I(10 μM), L-C_12_ (10 μM), and L-an (10 μM) for 7 days. (**E**) Enlargement of the red rectangle selected in D (scale bars = 5 mm). (**F**,**G**) Number and area of plaques calculated using Adobe Photoshop version 2020 (* *p* < 0.05, ** *p* < 0.01, *** *p* < 0.001 and **** *p* < 0.0001 compared to DMEM; ^#^ *p* < 0.05, ^##^ *p* < 0.01 and ^###^ *p* < 0.001 compared to Lycosin-I).

**Figure 4 toxins-15-00477-f004:**
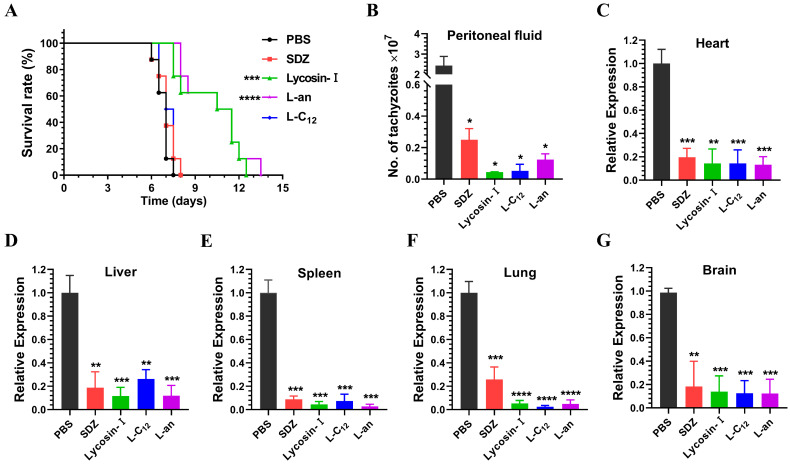
Anti-*T. gondii* of lipopeptides in vivo. Mice acutely infected with *T. gondii* were treated with PBS (negative control), SDZ (4 mg/kg, positive control), Lycosin-I (4 mg/kg), L-C_12_ (4 mg/kg) and L-an (4 mg/kg). (**A**) The survival time of mice was recorded for 15 days (*n* = 8 for each group; *** *p* < 0.001 and **** *p* < 0.0001 compared to PBS). (**B**–**G**) The number of tachyzoites in the peritoneal fluid of mice was directly counted using a blood cell counting plate, while the expression of SAG1 for tachyzoites was detected via qRT-PCR in the heart, liver, spleen, lung and brain of mice. Analysis was performed using the comparative threshold cycle method (2^−∆∆Ct^) (* *p* < 0.05, ** *p* < 0.01, *** *p* < 0.001 and **** *p* < 0.0001 compared to PBS).

**Figure 5 toxins-15-00477-f005:**
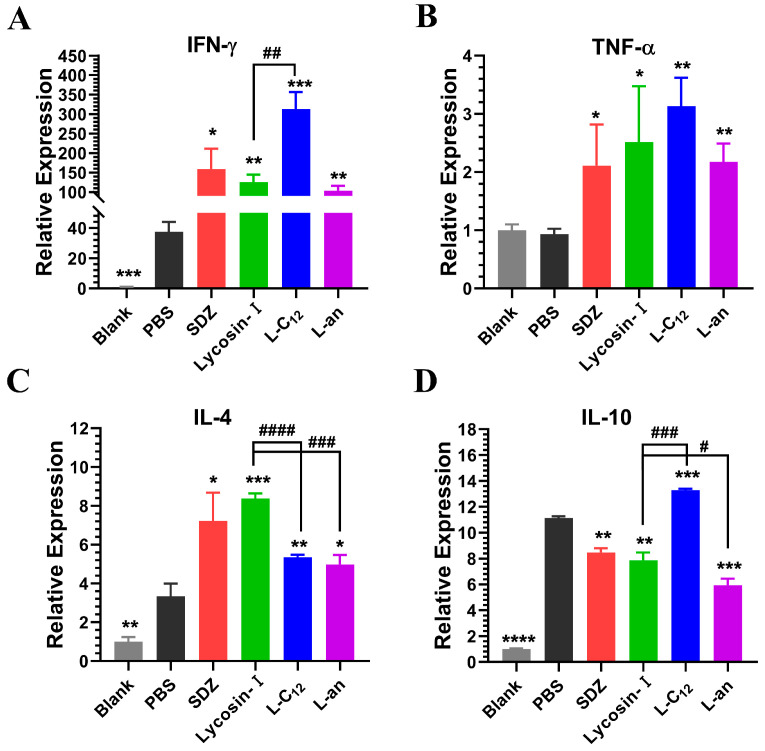
Expression of inflammatory factors in the spleen of mice. Mice acutely infected with *T. gondii* were treated with PBS (negative control), SDZ (4 mg/kg, positive control), Lycosin-I (4 mg/kg), L-C_12_ (4 mg/kg) and L-an (4 mg/kg). After 5 days of treatment, the expression level of IFN-γ (**A**), TNF-α (**B**), IL-4 (**C**) and IL-10 (**D**) in the spleen was determined via qRT-PCR. Analysis was performed using the comparative threshold cycle method (2^−∆∆Ct^) (* *p* < 0.05, ** *p* < 0.01, *** *p* < 0.001 and **** *p* < 0.0001 compared to PBS; ^#^ *p* < 0.05, ^##^ *p* < 0.01, ^###^ *p* < 0.001 and ^####^ *p* < 0.0001 compared to Lycosin-I).

**Table 1 toxins-15-00477-t001:** Reversed-phase high-performance liquid chromatography (RP-HPLC) with acetonitrile gradient. Detection wavelength: 280 nm; elution solution A: ddH_2_O (0.1% Trifluoroacetic acid (TFA)); B: acetonitrile (0.1% TFA).

Time (min)	Flow Rate (mL/min)	A%	B%
0	3	80	20
5	3	80	20
35	3	10	90
38	3	10	90
38.1	3	90	10
41	3	90	10

**Table 2 toxins-15-00477-t002:** Primers for qRT-PCR. F means forward primer. R indicates reverse primer.

Name of Primers	Seq of Primers
SAG1-F	5′-CGAGTATGTTTCCGAAGGCAGTGAG-3′
SAG1-R	5′-GCAGGTGACAACTTGATTGGCAAC-3′
β-Actin (*T. gondii*)-F	5′-GCTCTGGCTCCTAGCACCAT-3′
β-Actin (*T. gondii*)-R	5′-GCCACCGATCCACACAGAGT-3′
IL-4-F	5′-TACCAGGAGCCATATCCACGGATG-3′
IL-4-R	5′-TGTGGTGTTCTTCGTTGCTGTGAG-3′
IL-10-F	5′-AGAGAAGCATGGCCCAGAAATCAAG-3′
IL-10-R	5′-CTTCACCTGCTCCACTGCCTTG-3′
IFN-γ-F	5′-CTGGAGGAACTGGCAAAAGGATGG-3′
IFN-γ-R	5′-GACGCTTATGTTGTTGCTGATGGC-3′
TNF-α-F	5′-CACCACGCTCTTCTGTCTACTGAAC-3′
TNF-α-R	5′-CACACTGTCTTCTTGCCCTCCTAAC-3′
GAPDH (mouse)-F	5′-TGTTTCCTCGTCCCGTAGA-3′
GAPDH (mouse)-R	5′-ATCTCCACTTTGCCACTGC-3′

## Data Availability

The datasets used and/or analyzed during the current study are available from the corresponding author on reasonable request.

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
