# Peer review of "Anti-Toxoplasma gondii Effects of Lipopeptide Derivatives of Lycosin-I"

_toxins, 2023, doi:10.3390/toxins15080477_

Round 1

Reviewer 1 Report

This work presents a rational study about how the addition of a fatty chain in the peptide lycosin-I improves its anti-T. gondii activity. The paper is well written with interesting in vitro and in vivo studies, thus should be considered for publication. 

However, some questions were raised during paper analysis.

1- The authors evaluated the effect of Lycosin-I against extracellular parasites using trypan blue dye. Figures of treated parasites incubated with trypan blue are shown in Figure S1. Analyzing the images presented, it is hard to differentiate viable to non-viable parasites in some images. Please, improve the images so the difference is clear, or maybe, the authors should consider using another viability assay, such as propidium iodide.

2- For the invasion inhibition assay, the authors chose 10 µM that showed high toxicity for extracellular tachyzoites, killing around 80% of tachyzoites, thus it is not unexpected that that had caused a high decrease in parasite invasion. Besides 5 µM, which I think is appropriate; I suggest checking the effect of 2.5 µM.

3- for invasion assays, authors used SDZ 10µM as a positive control. 

SDZ targets dihydropteroate synthase, affecting parasite proliferation. It is not expected that this drug inhibit the parasite invasion, thus SDZ is not a good positive control for this kind of experiment or maybe mention that SDZ is not expected to inhibit parasite invasion. As an invasion-positive control authors could use EDTA or 1,10-phenanthroline

4-In relation to the antiproliferative assay, the concentration of SDZ used (10 µM) is not effective against Toxoplasma when used alone for 24h, however, a significant effect was observed (?). There are several works in the literature showing that SDZ acts in concentrations above 100µM, for example: 

-Harris et al, 1988 showed that 100µM did not inhibit tachyzoites proliferation.

-Doliwa et al, 2011 showed that SDZ inhibited RH and Me49 strains with IC50s of 260 μM for and 176 μM, respectively, after 72h-treatment in vitro.

Thus, I recommend that the authors check the effect of SDZ again.

5- The in vivo experiments showed that both SDZ and LC-12 treatments did not prolong mice survival compared to the PBS group. However, when tachyzoite load in peritoneal fluid and organs were investigated, LC-12 mice showed a low amount of parasite in peritoneal fluid and lungs, similar to those found for lycos and L-an. 

The authors discussed a possible toxic effect of LC-12 on mice. I believe that this experiment would improve the work. Although LC-12 promoted the IL-10 expression, that also promoted IFN-g and TNF-a expression. Is it toxicity or immune response imbalance? 

Is it possible to include a short experiment treating non-infected mice with LC-12?

6 - What was the route of compound administration in vivo? i.p? oral? This information should be provided.

7- In discussion session lines 266-270, the authors affirm that the modification of the α-amino terminus of the first lysine site of Lycosin-I improved anti-T. gondii activity, however, this is overstated. Analyzing the presented results, lycosin and LC-12 did not show a high difference in the IC50 after parasite extracellular treatment and activity after intracellular incubation. 

8- Lines 295-296: “These results showed that fatty acid modification enhanced the ability of Lycosin-I to inhibit tachyzoite invasion by reducing their motility, adhesion, and locomotion.” The authors did not evaluate parasite motility, adhesion, or locomotion after treatment. Thus, this is speculative. 

9-Lines 302-304: “We hypothesized that Lycosin-I, L-C12, and L-an could inhibit tachyzoite proliferation by appropriately promoting the host inflammatory response and inhibiting the release of proteins secreted by tachyzoites into the host cytoplasm.”

This is very speculative, I suggest removing this hypothesis, as it is not possible to affirm that the tested peptides inhibit the parasite secretion. The results shown in this work did not provide any evidence of this mechanism. 

One suggestion is to check if the treatment of HFF cells with peptides induces the expression of microbicidal mediators, maybe it is an effect on the host cell and not on the parasite, or authors could evaluate the parasite secretion after treatment. 

10 - A statement to confirm that all experimental protocols with mice were approved by the Institution or a committee is missing. This information should be included  

Minor:

The tachyzoite precipitate… 393, substitute precipitate for pellet. 

Author Response

Point 1: The authors evaluated the effect of Lycosin-I against extracellular parasites using trypan blue dye. Figures of treated parasites incubated with trypan blue are shown in Figure S1. Analyzing the images presented, it is hard to differentiate viable to non-viable parasites in some images. Please, improve the images so the difference is clear, or maybe, the authors should consider using another viability assay, such as propidium iodide.

Response 1: Thank you for your nice comments. We are sorry for the lack of clarity in the images due to the layout of our groups. We have enlarged the graphs in Figure S1. When the cells are damaged or dead, the trypan blue stain can penetrate through the degenerated cell membrane and stain the disintegrated DNA, so that the dead cells can be seen as swollen and blue under the light microscope; while the cell membrane of living cells is intact, the blue stain cannot penetrate the cell membrane of living cells, so that living cells can be observed as colourless and transparent with refractive properties under the light microscope. As you say, the propidium iodide is another validated assay for cell viability and we have applied it to other studies. In addition, we have also used RH-GFP (a fluorescent strain that expressesing GFP) to verify that the results are consistent.

Point 2: For the invasion inhibition assay, the authors chose 10 µM that showed high toxicity for extracellular tachyzoites, killing around 80% of tachyzoites, thus it is not unexpected that that had caused a high decrease in parasite invasion. Besides 5 µM, which I think is appropriate; I suggest checking the effect of 2.5 µM.

Response 2: Thank you very much for your professional comments. In our previous study we found that the parent peptide Lycosin-â…  also inhibited Toxoplasma tachyzoites invasion at lower concentrations (1.25 μM and 2.5 μM)[1]. Due to our poor consideration, we did not set a more comprehensive and reasonable concentration range when comparing the effects of these modified peptides with the parent peptide Lycosin-â…  on Toxoplasma gondii (T. gondii) invasion. Your suggestions are very helpful in improving our work. In fact, our research on these peptides does not end and we would like to further investigate the effect of these peptides on the chronic infection of T. gondii (bradyzoite cysts). We strongly endorse your suggestion and include it in our next study.

Point 3: for invasion assays, authors used SDZ 10µM as a positive control. 

SDZ targets dihydropteroate synthase, affecting parasite proliferation. It is not expected that this drug inhibit the parasite invasion, thus SDZ is not a good positive control for this kind of experiment or maybe mention that SDZ is not expected to inhibit parasite invasion. As an invasion-positive control authors could use EDTA or 1,10-phenanthroline.

Response 3: Thank you very much for your professional and valuable advices. Your suggestions are very helpful. We have subsequently found that SDZ does not directly kill T. gondii tachyzoites in vitro in other studies with trypan blue assay. As you say, SDZ targets dihydropteroate synthase and thus affects T. gondii proliferation rather than directly killing T. gondii, so the choice of SDZ as a positive control in the invasion assay is inappropriate. However, considering that these lipopeptides were obtained by modifying Lycosin-â… , which has been shown to have anti- T. gondii activity in our previous studies[1], and we mainly wanted to compare the anti- T. gondii activity of these modified peptides with the parent peptide Lycosin-â… , no additional other positive control was performed. We have added a description for this in the discussion (line 295-301). Your suggestion to use EDTA or 1,10-phenanthroline as a positive control for the invasion assay is excellent and we will use your valuable advice in other studies, thank you again.

Point 4: In relation to the antiproliferative assay, the concentration of SDZ used (10 µM) is not effective against Toxoplasma when used alone for 24h, however, a significant effect was observed (?). There are several works in the literature showing that SDZ acts in concentrations above 100µM, for example: 

-Harris et al, 1988 showed that 100µM did not inhibit tachyzoites proliferation.

-Doliwa et al, 2011 showed that SDZ inhibited RH and Me49 strains with IC50s of 260 μM for and 176 μM, respectively, after 72h-treatment in vitro.

Thus, I recommend that the authors check the effect of SDZ again.

Response 4: Thank you for your comment. As you said, we also considered that SDZ exerts its anti- T. gondii effect only at high concentrations, and its anti- T. gondii effect was also observed at t 400 µM in vitro and 100 mg/kg in vivo in our previous study[2]. In this study, we wanted to explore whether the anti- T. gondii effect of the peptide was better at the same low concentration as the SDZ, so we set the SDZ at the same concentration as the peptide in both in vitro and in vivo experiments. As you said, we also considered that SDZ exerts its anti- T. gondii effect only at high concentrations, and its anti- T. gondii effect was also observed at 400 µM in our previous study[1, 2]. In this study, we wanted to explore whether the anti- T. gondii effect of the peptide was better at the same low concentration as the SDZ, so we set the SDZ at the same concentration as the peptide in both in vitro and in vivo experiments. Interestingly, we have repeated the experiment many times and observed that SDZ inhibits T. gondii proliferation in host cells in vitro, especially in the plaque assay, where almost no significant plaques were observed.

Point 5: The in vivo experiments showed that both SDZ and LC-12 treatments did not prolong mice survival compared to the PBS group. However, when tachyzoite load in peritoneal fluid and organs were investigated, LC-12 mice showed a low amount of parasite in peritoneal fluid and lungs, similar to those found for lycos and L-an. 

The authors discussed a possible toxic effect of LC-12 on mice. I believe that this experiment would improve the work. Although LC-12 promoted the IL-10 expression, that also promoted IFN-g and TNF-a expression. Is it toxicity or immune response imbalance? 

Is it possible to include a short experiment treating non-infected mice with LC-12?

Response 5: Thank you very much for your professional and valuable advices. Your suggestions are of great help in improving our work. In fact, our research on these peptides does not end and we would like to further investigate the effect of these peptides on the chronic infection of T. gondii (bradyzoite cysts). In addition, we need some time to synthesize these peptides and we would like to complement our next work with in vivo toxicity experiments on these peptides if you allow us to do so.

Point 6: What was the route of compound administration in vivo? i.p? oral? This information should be provided.

Response 6: Thank you for your comments. We are sorry for the lack of details in my previous work and we have redescribed the survival assay in the part of materials and methods. As these peptides are small linear peptides that are easily degraded by pepsin, mice were treated with these peptides by intraperitoneal injection (in the same way as in mice infected with T. gondii) rather than oral gavage.

Point 7: In discussion session lines 266-270, the authors affirm that the modification of the α-amino terminus of the first lysine site of Lycosin-I improved anti-T. gondii activity, however, this is overstated. Analyzing the presented results, lycosin and LC-12 did not show a high difference in the IC50 after parasite extracellular treatment and activity after intracellular incubation. 

Response 7: Thank you for your professional advice. We have removed this inaccurate description. In fact, combining with the results of the trypan blue assay and the serum incubation, it shown that L-C12 retains the same anti-Toxoplasma activity as the parent peptide Lycosin-â…  and that its anti- T. gondii activity is not affected by the serum. What we want to express is that the α-amino terminus of the first lysine site of Lycosin-I (the N-terminus of Lycosin-I) was the better site for modification compared toother lysine sites.

Point 8: Lines 295-296: “These results showed that fatty acid modification enhanced the ability of Lycosin-I to inhibit tachyzoite invasion by reducing their motility, adhesion, and locomotion.” The authors did not evaluate parasite motility, adhesion, or locomotion after treatment. Thus, this is speculative. 

Response 8: Thank you for your professional comment. We have removed this inaccurate description.

Point 9: Lines 302-304: “We hypothesized that Lycosin-I, L-C12, and L-an could inhibit tachyzoite proliferation by appropriately promoting the host inflammatory response and inhibiting the release of proteins secreted by tachyzoites into the host cytoplasm.”

This is very speculative, I suggest removing this hypothesis, as it is not possible to affirm that the tested peptides inhibit the parasite secretion. The results shown in this work did not provide any evidence of this mechanism. 

One suggestion is to check if the treatment of HFF cells with peptides induces the expression of microbicidal mediators, maybe it is an effect on the host cell and not on the parasite, or authors could evaluate the parasite secretion after treatment. 

Response 9: Thank you very much for your professional and valuable advice. We have removed this inaccurate description. We strongly endorse your suggestion and include it in our next study.

Point 10: A statement to confirm that all experimental protocols with mice were approved by the Institution or a committee is missing. This information should be included. 

Response 10: Thank you for your comment. We are sorry for our carelessness and we have added animal ethics approval numbers (line 478).

Minor:

Point 11: The tachyzoite precipitate… 393, substitute precipitate for pellet. 

Response 11: Thank you for your nice comments. We have substituted precipitate for pellet.

REFERENCES

  1. Tang, Y. Q.; Hou, S. J.;  Li, X. Y.;  Wu, M. Q.;  Ma, B. B.;  Wang, Z.;  Jiang, J. Y.;  Deng, M. C.;  Duan, Z. G.;  Tang, X.;  Liu, Y.;  Wang, W. H.;  Han, X. Q.; Jiang, L. P., Anti-parasitic effect on Toxoplasma gondii induced by a spider peptide lycosin-I. Exp Parasitol 2019, 198, 17-25.
  2. Liu, Y.; Tang, Y. Q.;  Tang, X.;  Wu, M. Q.;  Hou, S. J.;  Liu, X. H.;  Li, J.;  Deng, M. C.;  Huang, S. Q.; Jiang, L. P., Anti-Toxoplasma gondii Effects of a Novel Spider Peptide XYP1 In Vitro and In Vivo (vol 9, 934, 2021). Biomedicines 2022, 10 (5).

Reviewer 2 Report

The authors have made a series of lipopeptide analogues of the spider toxin lycosin-I and tested their efficacy against the parasite Toxoplasma gondii. The strength of this paper is that a series of analogues have been designed and tested both in vitro and in vivo. The figure with the model of the lipoprotein is very helpful for understanding this manuscript.

However, several major revisions are required for this manuscript:

REVISIONS:

-       Make figures bigger, had to zoom in to look at the graphs in Fig 1C and the resolution was low

-       Would be informative to add a figure of normalized concentration-response curves (mortality as a percentage of PBS control) constructed from the bar graphs in Fig 1C so that the reader can see the difference in potency of the various analogues

-       What is the positive control used for the trypan blue assay?

-       How does lycosin-I and lipoprotein analogues compare to existing antiprotozoal drugs in the same assay? Authors should provide IC50s for comparison so that the reader can evaluate what improvement if any these peptides offer over existing drugs.

-       Authors should state in text (line 150) what host cell line is being used to evaluate cytotoxicity

-       Authors should provide an estimate of the selectivity index of their lipopeptides based on the IC50s for toxoplasma and host cell lines

-       In vitro and in vivo need to be italicized (e.g. line 205)

-       Authors need to include animal ethics approval numbers

-       Authors should provide a description of the RP-HPLC methods in 5.2 (e.g. column time, gradient)

-       The serum stability assay used by the authors does not reflect current practices or physiology. The authors have incubated their peptides at 4C in 10% FBS for 24 hours, where normally such assays are performed at physiological temperature with gentle agitation. The peptides are then analyzed by LC and MS to determine the half-life and stability over time. Calling this a serum stability assay is inaccurate and misleading.To measure serum stability, the experiment must be repeated in accordance with standard practice and a half-life calculated.

-       Survival assay is missing lots of critical information: What were the weights of the mice? How were the mice treated? Oral gavage/feed/intraperitoneal injection? Ethics approval protocol number? Were there any signs of pain or toxicity from the lipopeptides? What was the reasoning for choosing 4 mg/kg/day dose for seven days? Looking at other papers, 10-160 mg/kg doses of sulfadiazine for 10-17 days are used.

Quality of English is okay, some minor revision required.

Author Response

Point 1:     Make figures bigger, had to zoom in to look at the graphs in Fig 1C and the resolution was low.

Would be informative to add a figure of normalized concentration-response curves (mortality as a percentage of PBS control) constructed from the bar graphs in Fig 1C so that the reader can see the difference in potency of the various analogues

Response 1: Thank you very much for your professional and valuable advices. Your suggestions are very helpful. We have re-adjusted Figure 1 and added a figure of normalized concentration-response curves (mortality as a percentage of PBS control) constructed from the bar graphs in Fig 1C.

Point 2:  What is the positive control used for the trypan blue assay?

Response 2: In the trypan blue assay, considering that these lipopeptides were obtained by modifying Lycosin-â… , which has been shown to have anti-Toxoplasma activity in our previous studies[1], so instead of setting up other positive controls, we initially compared the anti-Toxoplasma gondii (T. gondii ) activity of these modified peptides with that of the parent peptide Lycosin-â… .

Point 3:  How does lycosin-I and lipoprotein analogues compare to existing antiprotozoal drugs in the same assay? Authors should provide IC50s for comparison so that the reader can evaluate what improvement if any these peptides offer over existing drugs.

Response 3: Thank you for your nice and professional comments. The main clinical anti-Toxoplasma drugs were sulfadiazine (SDZ) and pyrimethamine. In this study, we chose SDZ as a positive control. Looking at other papers, SDZ exerts its anti- T. gondii effect at high concentrations, and its anti- T. gondii activity was also observed at 400 µM in vitro and 100 mg/kg in vivo in our previous study[2]. In this study, we wanted to explore whether the anti- T. gondii activity of the peptides was better at the same low concentration as the SDZ, so we set the SDZ at the same concentration as the peptides in both in vitro and in vivo experiments. As you said, IC50s can help the reader to quickly assess the difference between these peptides and clinical drugs. However, due to our poor consideration, we did not set a more comprehensive and reasonable concentration range when testing the effect of these peptides on T. gondii proliferation, which makes it impossible to calculate the IC50s of these peptides on T. gondii proliferation. Your advice will be of great help to us in our future research, thank you again.

Point 4:  Authors should state in text (line 150) what host cell line is being used to evaluate cytotoxicity.

Response 4: Thank you for your nice and professional comments. We are sorry for the caralessness. We have stated in text (line 151) that Human foreskin fibroblasts (HFFs) were used to evaluate cytotoxicity.

Point 5:   Authors should provide an estimate of the selectivity index of their lipopeptides based on the IC50s for toxoplasma and host cell lines

Response 5: Thank you for your nice comments. Due to our poor consideration, we did not set a more comprehensive and reasonable concentration range when testing the effect of these peptides on T. gondii proliferation, which makes it impossible to calculate the IC50 of these peptides on T. gondii proliferation, so no selectivity index is available based on the current experiments. Your suggestion is excellent and we will use your valuable advice in other studies, thank you again.

Point 6:  In vitro and in vivo need to be italicized (e.g. line 205)

Response 6: Thank you for pointing out our mistakes. We have italicized ‘in vitro’ and ‘in vivo’ in the full text.

Point 7:  Authors need to include animal ethics approval numbers.

Response 7: Thank you for your comment. We are sorry for our carelessness and we have added animal ethics approval numbers (line 478).

Point 8:  Authors should provide a description of the RP-HPLC methods in 5.2 (e.g. column time, gradient)

Response 8: Thank you for your nice and professional comments. We have added a table of RP-HPLC with acetonitrile gradient in 5.2.

Point 9:  The serum stability assay used by the authors does not reflect current practices or physiology. The authors have incubated their peptides at 4C in 10% FBS for 24 hours, where normally such assays are performed at physiological temperature with gentle agitation. The peptides are then analyzed by LC and MS to determine the half-life and stability over time. Calling this a serum stability assay is inaccurate and misleading. To measure serum stability, the experiment must be repeated in accordance with standard practice and a half-life calculated.

Response 9: Thank you for your professional comments. We are sorry for the error in description. Actually, we incubated peptides at 37 ℃ in 10% FBS for 24 hours following the instruction of Dr. Zhang[3]. The serum stability of Lycosin-I and L-C12 have been analyzed by MS in previous study[3]. In this study we wanted to explore whether the anti-Toxoplasma activity of these peptides was influenced by the serum. Interestingly, we found that although mass spectrometry confirmed the degradation of Lycosin-Ⅰ into smaller molecules of peptide after 24 hours of serum incubation, its anti- Toxoplasma activity in vitro was only slightly reduced, rather than completely lost, which suggests that although Lycosin-I was degraded into smaller peptides, the key amino acid sequences of anti-T. gondii were still retained in the small peptide, maybe we could optimize the specific activity of peptides by truncating them.

Point 10:  Survival assay is missing lots of critical information: What were the weights of the mice? How were the mice treated? Oral gavage/feed/intraperitoneal injection? Ethics approval protocol number? Were there any signs of pain or toxicity from the lipopeptides? What was the reasoning for choosing 4 mg/kg/day dose for seven days? Looking at other papers, 10-160 mg/kg doses of sulfadiazine for 10-17 days are used.

Response 10: Thank you for your nice and professional comments. We are sorry for the lack of details in my previous work and we have redescribed the survival assay in the part of materials and methods. As these peptides are small linear peptides that are easily degraded by pepsin, mice were treated with these peptides by intraperitoneal injection (in the same way as in mice infected with Toxoplasma gondii) rather than oral gavage.

As you said, SDZ exerts its anti-Toxoplasma effect at high concentrations in other papers, and its anti-Toxoplasma effect was also observed at 400 µM in vitro and 100 mg/kg in vivo in our previous study[2]. In this study, we wanted to explore whether the anti-Toxoplasma effect of the peptides was better at the same low concentration as the SDZ, so we set the SDZ at the same concentration as the peptides in both in vitro and in vivo experiments.

Looking at other papers, most mice with acute Toxoplasma infection are treated with drugs for 5-10 days. Considering mice acutely infected with T. gondii (1103 tachyzoites per mouse) usually start to die on the fifth day after infection and all untreated mice will die by 7-8 days, we chose 7 days of drug treatment to observe the survival of mice infected with T. gondii.

REFERENCES

  1. Tang, Y. Q.; Hou, S. J.;  Li, X. Y.;  Wu, M. Q.;  Ma, B. B.;  Wang, Z.;  Jiang, J. Y.;  Deng, M. C.;  Duan, Z. G.;  Tang, X.;  Liu, Y.;  Wang, W. H.;  Han, X. Q.; Jiang, L. P., Anti-parasitic effect on Toxoplasma gondii induced by a spider peptide lycosin-I. Exp Parasitol 2019, 198, 17-25.
  2. Liu, Y.; Tang, Y. Q.;  Tang, X.;  Wu, M. Q.;  Hou, S. J.;  Liu, X. H.;  Li, J.;  Deng, M. C.;  Huang, S. Q.; Jiang, L. P., Anti-Toxoplasma gondii Effects of a Novel Spider Peptide XYP1 In Vitro and In Vivo (vol 9, 934, 2021). Biomedicines 2022, 10 (5).
  3. Zhang, P.; Jian, C.;  Jian, S.;  Zhang, Q.;  Sun, X.;  Nie, L.;  Liu, B.;  Li, F.;  Li, J.;  Liu, M.;  Liang, S.;  Zeng, Y.; Liu, Z., Position Effect of Fatty Acid Modification on the Cytotoxicity and Antimetastasis Potential of the Cytotoxic Peptide Lycosin-I. J Med Chem 2019, 62 (24), 11108-11118.

Round 2

Reviewer 1 Report

I am satisfied with the manuscript modifications and authors' responses.

Author Response

Thank you again for reviewing the article and for your valuable comments!